# Physiological and transcriptomic analyses reveal the cadmium tolerance mechanism of *Miscanthus lutarioriparia*

Jia Wang[1,2,3], Xinyu Liu[2], Yiran Chen[2], Feng lin Zhu[1,2], Jiajing Sheng[3]*, Ying Diao[4]*

1 Joint National-Local Engineering Research Centre for Safe and Precise Coal Mining, Anhui University of Science and Technology, Huainan, P. R. China, 2 Key Laboratory of Industrial Dust Prevention and Control & Occupational Safety and Health of the Ministry of Education, Anhui University of Science and Technology, Huainan, P. R. China, 3 State Key Laboratory of Hybrid Rice, Hubei Lotus Engineering Center, College of Life Sciences, Wuhan University, Wuhan, P. R. China, 4 School of life science and technology, Wuhan Polytechnic University, Wuhan, P. R. China

* sjiajing@ntu.edu.cn (JS); ydiao@whu.edu.cn (YD)

**Data Availability Statement:** All data are available from the NCBI database (SRA data: PRJNA733881). https://www.ncbi.nlm.nih.gov/bioproject/?term=PRJNA733881.

## Abstract

*Miscanthus lutarioriparia* is a promising energy crop that is used for abandoned mine soil phytoremediation because of its high biomass yield and strong tolerance to heavy metals. However, the biological mechanism of heavy metal resistance is limited, especially for applications in the soil restoration of mining areas. Here, through the investigation of soil cadmium(Cd) in different mining areas and soil potted under Cd stress, the adsorption capacity of *Miscanthus lutarioriparia* was analyzed. The physiological and transcriptional effects of Cd stress on *M. lutarioriparia* leaves and roots under hydroponic conditions were analyzed. The results showed that *M. lutarioriparia* could reduce the Cd content in mining soil by 29.82%. Moreover, different Cd varieties have different Cd adsorption capacities in soils with higher Cd concentration. The highest cadmium concentrations in the aboveground and belowground parts of the plants were 185.65 mg/kg and 186.8 mg/kg, respectively. The total chlorophyll content, superoxide dismutase and catalase activities all showed a trend of increasing first and then decreasing. In total, 24,372 differentially expressed genes were obtained, including 7735 unique to leaves, 7725 unique to roots, and 8912 unique to leaves and roots, which showed differences in gene expression between leaves and roots. These genes were predominantly involved in plant hormone signal transduction, glutathione metabolism, flavonoid biosynthesis, ABC transporters, photosynthesis and the metal ion transport pathway. In addition, the number of upregulated genes was greater than the number of downregulated genes at different stress intervals, which indicated that *M. lutarioriparia* adapted to Cd stress mainly through positive regulation. These results lay a solid foundation for breeding excellent Cd resistant *M. lutarioriparia* and other plants. The results also have an important theoretical significance for further understanding the detoxification mechanism of Cd stress and the remediation of heavy metal pollution in mining soil.

**Funding:** This research was funded by Independent Research fund of Key Laboratory of Industrial Dust Prevention and Control & Occupational Health and Safety (Anhui University of Science and Technology) (NO.EK20202002), the Independent Research fund of Joint National-Local Engineering Research Centre for Safe and Precise Coal Mining (Anhui University of Science and Technology) (NO.EC2021007) and the Postdoctoral Science Foundation of China (2019M662719)." The funders had no role in study design, data collection and analysis, decision to publish, or preparation of the manuscript.

**Competing interests:** NO authors have competing interests.

## Introduction

The exploitation and utilization of mineral resources are an important source of heavy metal contamination in environmental systems [1]. The tailings and waste areas of all kinds of metal mines contain high concentrations of toxic heavy metals. The release and migration of heavy metals elements have caused serious pollution to the nearby soils and ecological environments and pose a great threat to people's quality of life and health [2–4]. Thus, how to remediate soil heavy metal contamination has always been a highly valued problem for the government and society. Phytoremediation methods have been widely used and studied because of their simple operation, low cost, obvious remediation effect, strong pertinence and lack of risk of secondary pollution [2,5,6]. These plants commonly exhibit fast growth, high biological yield, developed roots and a high bioaccumulation coefficient and can be used for phytoremediation [7,8]. Phytoremediation technology cannot only remediate the heavy metal pollution in soil but also prevent soil erosion and help to restore vegetation to improve the ecological environment of mining areas [9–11]. The efficiency of phytoremediation technology for treating heavy metal polluted soil mostly depends on plant biomass and the enrichment capacity for metal elements. At present, more than 700 kinds of heavy metal hyperaccumulators have been reported, including more than 20 species of cadmium (Cd) hyperaccumulators [7,12]. However, these hyperaccumulators generally have the characteristics of small biomasses, and limited tolerance and adaptability to heavy metals, which seriously limites the development of phytoremediation technology for soil remediation [2,7,13]. Therefore, the screening and breeding of plants with resistance to stress, wide adaptability, heavy metal resistance and strong adsorption capacity are critical problems in bioremediation in mining areas.

*Miscanthus lutarioriparia* is a perennial herbaceous C4 plant, belonging to the *Miscanthus* genus specific to China[14]. *M. lutarioriparia* has become a type of cellulosic energy crop with great development prospects because of its rapid growth, high yield, low input, easy management, wide distribution, and high resistance [15]. It is widely distributed in lakes, rivers and shoals in the Yangtze River of China and can quickly form dominant natural communities in river tidal flats, wastelands, and abandoned mining areas [14,15]. In addition, *M. lutarioriparia* reduced the content of heavy metals in polluted soil via root fixation [16,17]. The accumulation rates of Cd and plumbum (Pb) in *Miscanthus* vulgaris are positively correlated with soil heavy metal pollution, but the adaptations and adsorption capacities of these plants also greatly differ [17,18]. At present, the plants of *Miscanthus* (*Miscanthus×giganteus*, *Miscanthus sinensis*, *Miscanthus lutarioriparia* and *Miscanthus floridulus*) are used for the remediation of soil heavy metal pollution [16,17,19]. In Europe and America, *Miscanthus×giganteus* has been widely considered a heavy metal phytoremediation plant [19–22]. In China, there are dominant communities of *Miscanthus* in many mining areas, and the heavy metal content of soil with *Miscanthus* in the same mining area is significantly lower than that of soil without *Miscanthus* [16,23]. For example, in a tailing area, the contents of Pb and Cd in the soil without *M. floridulus* growth were 2630 mg/kg and 31.7 mg/kg, respectively. The contents of Pb and Cd decreased to 870 mg/kg and 9.1 mg/kg respectively, in the soil, with 16.7% distribution coverage of *M. floridulus* [23]. Preliminary studies have shown that the soil heavy metal content of *M. lutarioriparia*-covered land was only 24.72% of that of the bare land in the same lead-zinc mining area, which revealed that *M. lutarioriparia* significantly reduced the soil Pb content in the mining area [16]. Thus, *M. lutarioriparia* is also considered to be an excellent crop that can be used for bioremediation in mining areas. However, research on the restoration of soil heavy metal pollution in mining areas for *M. lutarioriparia*, has focused mainly on its morphology, physiology, and enrichment capacity. There is almost no research on its heavy metal tolerance mechanism.

Cd is a toxic heavy metal pollutant in soil, and its excessive accumulation can seriously affect the growth and development of plants and lead to death [24–26]. The adaptation by which plants adapt to heavy metal stress is complex and involves cell wall precipitation, the use of metal chelators (phytochelatins and metallothioneins), and the use of antioxidant systems [8,27–29]. Cell wall precipitation occurs mainly through carrier proteins transporting heavy metal ions to reduce heavy metal concentrations in cells, these proteins include the heavy metal ATPase (HMA) family proteins, ATP-binding cassette (ABC) transporter family proteins, and metal tolerance protein (MTP) family proteins [28,30]. There are few studies on the biological mechanism of *Miscanthu*s tolerance to heavy metal stress. It is generally believed that root metabolism, the antioxidant system and beneficial microorganisms in the rhizosphere are the major reasons why *Miscanthus* is tolerant to heavy metal stress. For example, the expression levels of the ATP binding cassette subfamily B member 1 (ABCB1) and ABCG2 genes in leaves and roots were significantly upregulated under Pb stress in *M. lutarioriparia* [16]. In recent years, transcriptomics has been widely used in research to reveal the molecular mechanism underlying the plant response to Cd stress [31].

In this study, through analysis of the Cd content in the land covered by *M. lutarioriparia* in different mining areas, we evaluated the ability of *M. lutarioriparia* to remediate soil Cd pollution. In addition, through the analysis of Cd content, antioxidant enzyme activity, chlorophyll content, and transcriptome sequencing of *M. lutarioriparia* leaves and roots under Cd stress, the physiological characteristics and molecular mechanisms of the genes involved were revealed. These results increase our understanding of the detoxification mechanism of Cd in *M. lutarioriparia* and identify candidate plants for phytoremediation. This study also lays a good foundation for the breeding of *M. lutarioriparia* and provides candidate genes for the genetic improvement of other Cd phytoremediation plants.

## Materials and methods

### Planting and treatment of experimental material

The experimental materials were obtained from the *Miscanthus* Resource Garden of Wuhan University and included MG068, ML004, ML042, ML325, ML104, ML123, ML505, AG002, ML008 and ML304, which were obtained from several mining areas and other areas (e.g., MG068 and AG002) (S1 Table). All materials were cultivated in the same flower pots with similar growing rhizomes. The soil used for pot cultivation was a uniform nutrient soil (pH 7.2, organic matter 15.2 g/kg). In addition, the Cd concentration in the soil was increased to 150 mg/kg by adding cadmium nitrate for the stress experiment. After soil treatment, the mixture was stirred and put into a pot for 20 days before use. All the experimental materials were planted in the treated soil, with 9 plants of each material. After 60 days of normal growing in glasshouses (day/night temperature 28°C/25°C, light/darkness duration 16 h/8 h, and humidity 60%), the aboveground and underground parts were removed to determine the Cd content.

In addition, to investigate the initial mechanism of adaptation to Cd stress, the physiological and transcriptomic effects of Cd stress on *M. lutarioriparia* (Ml004) leaves and roots were analyzed. The rhizomes of ML004, which are considered to have strong resistance to heavy metals, were cultured in a greenhouse. After 10 days, 25 plants with similar growth were transplanted into plastic pots for hydroponic growth. Quartz sand, water and Hoagland nutrient solution were added, and the plants were grown normally in a greenhouse (day/night temperature 28°C/25°C, light/darkness duration 16 h/8 h, and humidity 60%). After 20 days, cadmium nitrate solution was added to the hydroponic tank to a concentration of 18 mg/L. Root and leaf samples were collected at 0, 1, 2 and 4 days after Cd treatment. All the samples were first washed three times with deionized water, quickly put into sample tubes, frozen with liquid

nitrogen and stored in a -80˚C freezer for treatments. Each sampling time was between 9:00 am and 10:00 am. Three biological replicates were used for each experiment in this study.

## Detection of Cd content in Miscanthus lutarioriparia

The Cd content of the plants was determined by nitric acid-hydrogen peroxide digestion and plasma–mass spectrometry (ICP–MS, Agilent 7900) [32]. Briefly, 0.2 g of the sample was collected, put into a digestive juice (5 ml of nitric acid and 2 ml of hydrogen peroxide), digested thoroughly, diluted to 50 ml, filtered, and the content of each element was measured on a machine. To ensure the accuracy of the test data, parallel alignment and QC quality control samples were used for quality control of the raw data, which confirmed that the relative difference was less than 10%, and that the quality was acceptable (allowable relative difference $< 4\%$). In addition, the recovery rate ($>97\%$) of the instrument was also analyzed during the testing process. The translocation factor was calculated by the following equation: TF = Cd concentration in the aboveground part / Cd concentration in the belowground part.

## Detection of Cd content in soil

In total, 96 topsoil samples were obtained in the mining areas, including 48 samples of *M. lutarioriparius* growing point soil and 48 bare soil samples, respectively. Each soil sample was collected with a stainless steel shovel (0–20 cm depth) and kept in a sampling bag with 3 subsamples per sampling site for mixing. The soil samples were thoroughly mixed and air-dried to remove impurities and stored for subsequent experiments. A 0.2 g soil mixture was used for digestion with HCl-HNO3-HF-H2O2 (1:4:1:1), and the inductively coupled ICP–MS was used for Cd content detection[32].

## Chlorophyll content detection

Chlorophyll content was measured by a spectrophotometric method after extraction with 95% ethanol [33]. Briefly, 0.2 g leaf samples (usually the middle part of the fifth leaf) were first rinsed with distilled water, cut into pieces, placed in a stoppered test tube, added to 10 ml of 95% ethanol solution, shaken well and stored away from light until the green leaves turned completely white. The absorbance of the supernatant at 665 and 649 nm was recorded. The chlorophyll contene was calculated according to a previously described method [33].

## CAT and SOD activity detection

Antioxidant enzyme extraction and enzyme activity determination were performed according to previous methods [34,35] with some modifications. Briefly, fresh tissue (0.2 g) was minced in a prechilled mortar, and 5 ml of sodium phosphate buffer (pH 7.8) was added for thorough grinding (the process was performed on ice). The ground mixture was collected and centrifuged (4˚C, 12000 g) for 20 min, and the supernatant was stored in a refrigerator for subsequent determination of various enzyme activities. The nitro blue tetrazolium (NBT) photochemical reduction method and the UV-VIS spectrophotometry method were used to determine the activities of SOD and CAT in the samples, respectively [34,35].

## Transcriptome sequencing and analysis

A total of 24 samples were used for transcriptome sequencing. A plant RNA extraction kit (Tian Gen) was used to extract total RNA. Qualified RNA was used for library construction for Illumina HiSeq (HiSeq 4000 SBS Kit (300 cycles)) sequencing. Fastp software (version 0.19.5), SeqPrep (https://github.com/jstjohn/SeqPrep) and Sickle (https://github.com/najoshi/sickle)

software were used for quality control of the raw sequencing data. After quality control, the original data, namely, the clean data (reads), were aligned with the *M. lutarioriparia* genome via Bowtie2 software (version 2.4.1). Then, StringTie (version 2.1.2) was used for assembly and comparison with the *M. lutarioriparia* genome annotation information. The expression levels of genes were calculated by RSEM software (version 1.3.3). Principal components analysis (PCA) and hierarchical clustering (Pearson, Euclidean) methods were used to evaluate the transcriptome similarity and repeatability between samples. Differentially expressed genes (DEG) between samples were screened by DESeq2 software (version 1.24.0, screening criteria FDR < 0.05 and |log2FC|≧1). BLAST+ (version 2.9.0) was used to compare the functional annotations of genes/transcripts in public databases to obtain corresponding annotation information. Besides, we used a Venn diagram to analyze the DEGs among the samples. Bioinformatics analysis of the DEGs was performed, such as Gene Ontology (GO) classification (Goatools software, Version 0.6.5), KEGG pathway enrichment (KOBAS software, Version 2.1.1), and gene set enrichment analyses (GSEA, Version 3.0). All data are deposited in the supporting information, and the datasets generated during the study are available in the NCBI (SRA data: PRJNA733881).

### Validation of quantitative real-time PCR (RT–qPCR)

To test the transcriptome data, we randomly selected 8 genes with significant differences in expression and verified the expression levels of the genes in the samples by RT–qPCR. Primer Premier 6.0 software was used to design the primer sequences (S2 Table). The eIF4a gene was used as an internal reference gene. RT–qPCR was performed using the StepOne plusTM Real-Time PCR system. The experimental data were analyzed by the $2^{-\Delta\Delta CT}$ method.

### Data analysis

Differences in the Cd content, chlorophyll content, CAT activity, and SOD activity were detected via one-way analysis of variance (ANOVA). Statistical analysis was performed using SPSS v.11 (SPSS Inc., USA), and $P<0.05$ was considered to indicate statistical significance. All the data are presented as the mean±standard deviation (SD).

## Results

### Cd detection in soil in the mining area

The contents of Cd in soil-covered land with *M. lutarioriparia* present and bare land were tested in 8 different mining areas. The results showed that *M. lutarioriparia* could significantly reduce the soil Cd content in its growth environment (Fig 1). In general, the Cd content in the bare soil was 3–6 times greater than that in the soil covered by *M. lutarioriparius*. In other words, *M. lutarioriparia* reduced the Cd content in mining soil by 29.82% - 82.87%. Thus, *M. lutarioriparia* had a strong tolerance and adsorption of Cd and could be used for mining soil heavy metal bioremediation.

### Cd content of M. lutarioriparia under the long-term Cd stress

The Cd contents in the aboveground and belowground parts of the 10 plants were different (Table 1). Among them, the ML104 aboveground and belowground Cd contents were the highest, with values of 185.65±15.54 mg/kg and 186.80±14.97 mg/kg, respectively. The highest transfer factor was found for ML004 (1.06), followed by ML104 (0.99) and ML008 (0.87). The results showed that the Cd adsorption capacity of *M. lutarioriparia* was strong, but Cd transfer

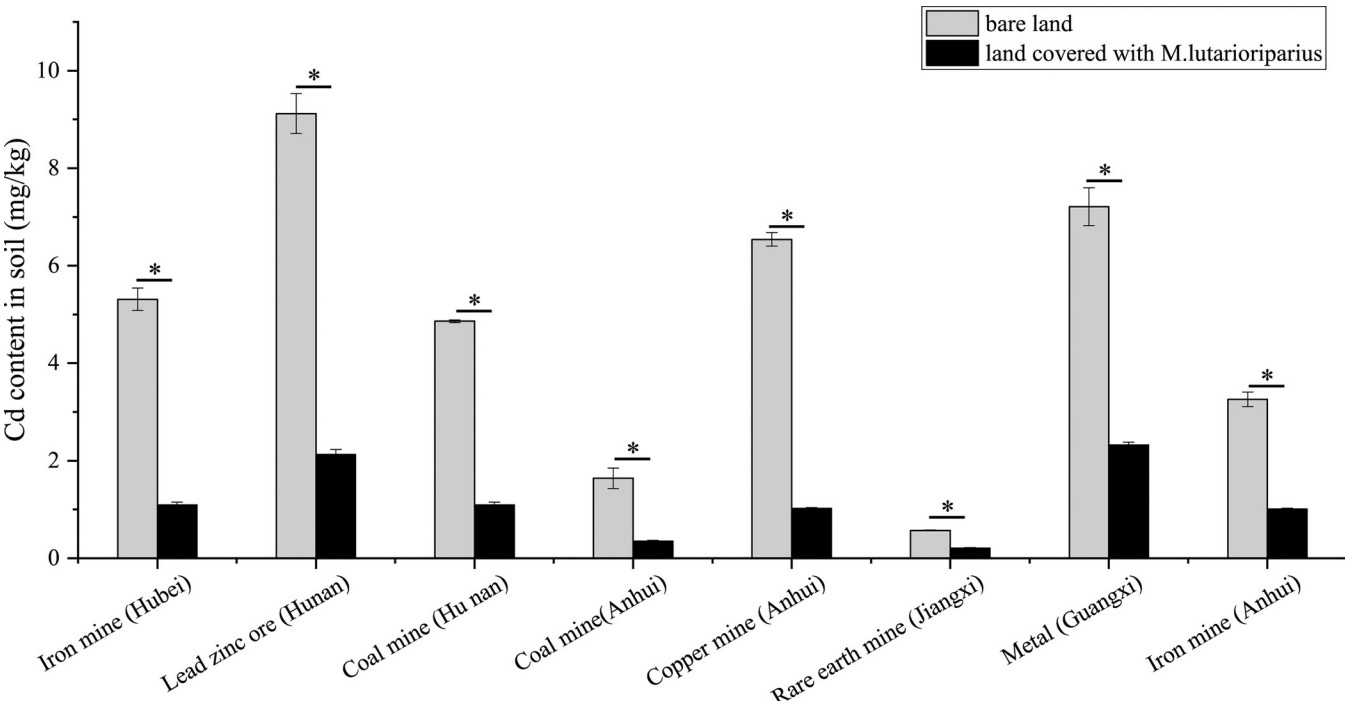

**Fig 1. Cd determination in soil in different mining areas.** The data are presented as the mean±standard deviation (SD), n = 6. * *P < 0.01 and *** p < 0.001 (one-way ANOVA).

was common. Therefore, *M. lutarioriparia* is mainly absorbed and fixed by the root system to reduce the Cd content in the soil.

## The initial response of M. lutarioriparia to Cd stress

To test the initial response of *M. lutarioriparia* to Cd treatment, the Cd content of leaves and roots was measured at different time points under Cd stress. The results displayed that the accumulation of Cd in leaves and roots increased with treatment time (Fig 2A). In addition,

**Table 1. Cd content of *Miscanthus lutarioriparia* under Cd stress.**

| Name | aboveground part (mg/kg) | belowground part (mg/kg) | Translocation factor |
|---|---|---|---|
| MG068 | 89.62±11.68 [a, B] | 135.26±5.34 [b, ABCD] | 0.66±0.03 [AB] |
| ML004 | 151.38±10.26 [a, E] | 161.97±9.68 [a, DE] | 1.06±0.04 [E] |
| ML042 | 127.29±13.64 [a, D] | 152.51±13.34 [b, BCD] | 0.83±0.03 [BCD] |
| ML325 | 106.31±15.21 [a, BCD] | 154.13±18.36 [b, CD] | 0.79±0.02 [BC] |
| ML104 | 185.65±6.89 [a, F] | 186.80±15.64 [a, E] | 0.99±0.02 [DE] |
| ML123 | 104.52±10.26 [a, BC] | 128.64±6.58 [b, ABC] | 0.81±0.04 [BCD] |
| ML505 | 113.92±11.64 [a, CD] | 159.64±19.64 [b, DE] | 0.71±0.02 [BC] |
| AG002 | 65.98±9.12 [a, A] | 131.34±10.34 [b, ABC] | 0.5±0.05 [A] |
| ML008 | 108.67±8.64 [a, BCD] | 124.67±24.36 [b, AB] | 0.87±0.02 [BCDE] |
| ML304 | 89.37±4.68 [a, B] | 112.69±9.98 [b, A] | 0.79±0.02 [BC] |

Data are presented as the mean±standard deviation (SD), n = 8, and different letters indicate significant differences, P<0.05 (one-way ANOVA). Lower-case letters indicate differences in data between the above-ground and underground parts of the plant (same line). Capital letters indicate differences between above-ground or underground parts of different plants (same column).

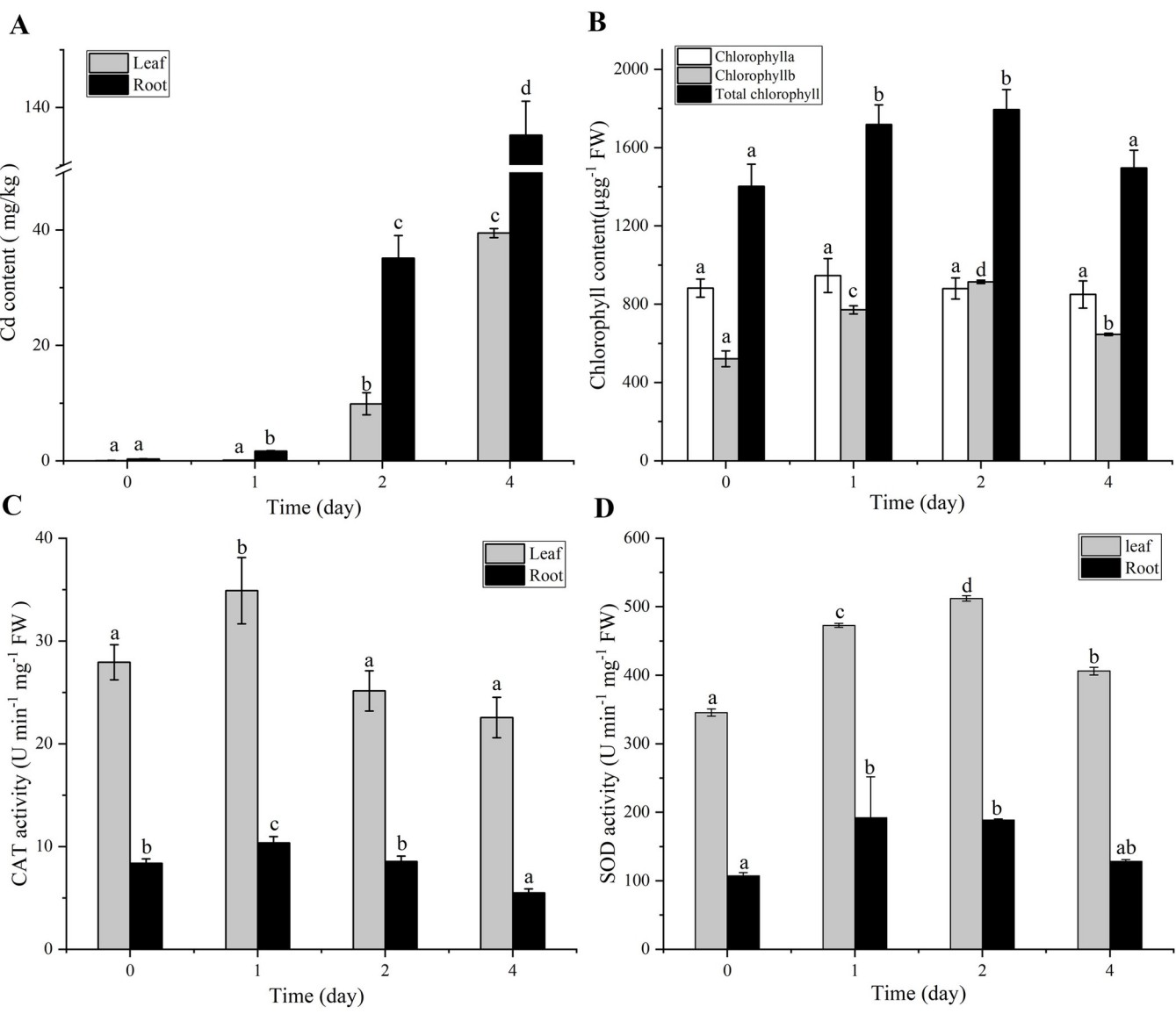

**Fig 2. The Cd content, chlorophyll content, and SOD and CAT activities changes under Cd stress in *M. lutarioriparia*.** (a) Cd content changes, (b) chlorophyll content changes, (c) CAT activity changes, and (d) SOD activity changes. The data are presented as the mean±standard deviation (SD), n = 3, and different letters indicate significant differences, P<0.05 (one-way ANOVA). One enzyme unit (U) was equal to the amount used to inhibit 50% of the photochemical reduction of NBT, and a decrease in absorbance of 0.01 at OD240 nm minute−1 was considered to indicate 1 U of CAT activity. FW is the fresh weight of leaves and roots.

the accumulation rate and amount of Cd in roots were significantly greater than those in leaves, which also indicated that the roots were used to adsorb and enrich Cd from *M. lutarioriparia*.

To analyze the toxicity of Cd to *M. lutarioriparia*, the chlorophyll content and antioxidant enzyme activity under normal conditions and after Cd treatment for 1, 2 and 4 days were tested. The contents of chlorophyll b and total chlorophyll in the *M. lutarioriparia* leaves showed an overall trend of first increasing and then decreasing with continued treatment time, and the maximum appeared on the second day (Fig 2B). However, the chlorophyll a content did not significantly change under Cd stress. Thus, *M. lutarioriparia* mainly adjusts the content of chlorophyll b to adapt to Cd stress.

The CAT and SOD activities showed the same changing trend in the leaves and roots, with both first increasing and then decreasing (Fig 2C and 2D). However, the greatest value of CAT activity occurred on the first day after the Cd treatment, while the greatest value of SOD activity occurred on the second day. In addition, the activities of CAT and SOD enzymes in leaves were significantly greater than those in the roots.

## Transcriptome assembly and annotation

To elucidate the initial response mechanism of *M. lutarioriparia* to Cd stress at the transcriptional level, transcriptome sequencing of leaves and roots was performed in the Cd treatment group (1, 2, 4 days) and the normal group (0 days). A total of 166.15 Gb of clean data were collected from 24 samples (each sample was above 6.92 Gb), and the percentage of Q30 was greater than 92.47% (S3 Table). These clean reads were compared with the *M. lutarioriparia* genome, and the mapping rate exceeded 86.55% (86.55%~92.29%) (S3 Table). Meanwhile, PCA revealed that 24 samples were divided into 8 categories according to the original sampling group (S1A Fig). These results showed that the clean read data between samples were comparable, and the biological reproducibility between samples was acceptable, which meet the requirements of subsequent analysis. After assembly, 81,142 genes were obtained from these clean reads, 12,814 of which were new genes. In addition, functional annotation of all genes was performed through comparison with public databases by BlastX with a cutoff value <le-5 (S1B Fig and S4 Table).

## Analysis of DEGs in response to Cd stress

With DESeq2 software analysis, a total of 24372 DEGs were obtained, including 7735 unique to leaves, 7725 unique to roots, and 8912 unique to both leaves and roots (Fig 3A and 3B, S1C Fig, S5 and S6 Tables). Compared with the control group, the more DEGs were detected with increasing of treatment time. Among them, L0_vs_L4 (13277) and R0_vs_R4 (11507) had the largest number of DEGs in the leaves and roots, respectively. Venn analysis revealed that most of the DEGs existed only at specific stages, but some DEGs (674 in L, 9 in R) continued to participate in the regulation under Cd stress (Fig 3C and 3D). In addition, Cd_1_vs_Cd_4 had the largest number of DEGs in roots (15190) and leaves (14813) among the different Cd stress stages, among which 1444 and 545 genes were involved in the response to Cd stress at different stress stages, respectively. Further analysis of the DEGs in all comparison groups in the leaves and roots revealed 211 DEGs in the leaves. In contrast, no DEGs in any of the comparison groups were found in the roots (S1D Fig). This showed that the regulatory mechanism in roots was more complicated and changeable than that in leaves. In addition, there were more upregulated genes than downregulated genes at different stress intervals, which indicated that *M. lutarioriparia* adapted to Cd stress mainly through positive regulation. In addition,we analyzed the expression levels of 8 randomly selected DEGs using qRT–PCR. The results were highly similar to the RNA-seq data, which indicated that the RNA-seq data were very reliable (S7 Table).

To investigate the biological functions of these DEGs, GO and KEGG enrichment analyses were used. KEGG enrichment revealed showed that the DEGs in the leaves were mostly enriched in photosynthesis, mitogen-activated protein kinase signaling pathway-plant (MAPK), glutathione metabolism, flavonoid biosynthesis, phenylalanine metabolism, starch and sucrose metabolism, and nitrogen metabolism (Fig 4A and 4B). In addition, The GO enrichment results showed that the DEGs in leaves were mostly enriched in metal ion transport, metal ion transmembrane activity, UDP-glucosyltransferase activity, glucosyltransferase activity, phenylalanine ammonia-lyase activity, response to oxygen-containing compounds, NAD(P)H oxidase activity, response to abiotic stimuli, etc. (S2A and S2B Fig).

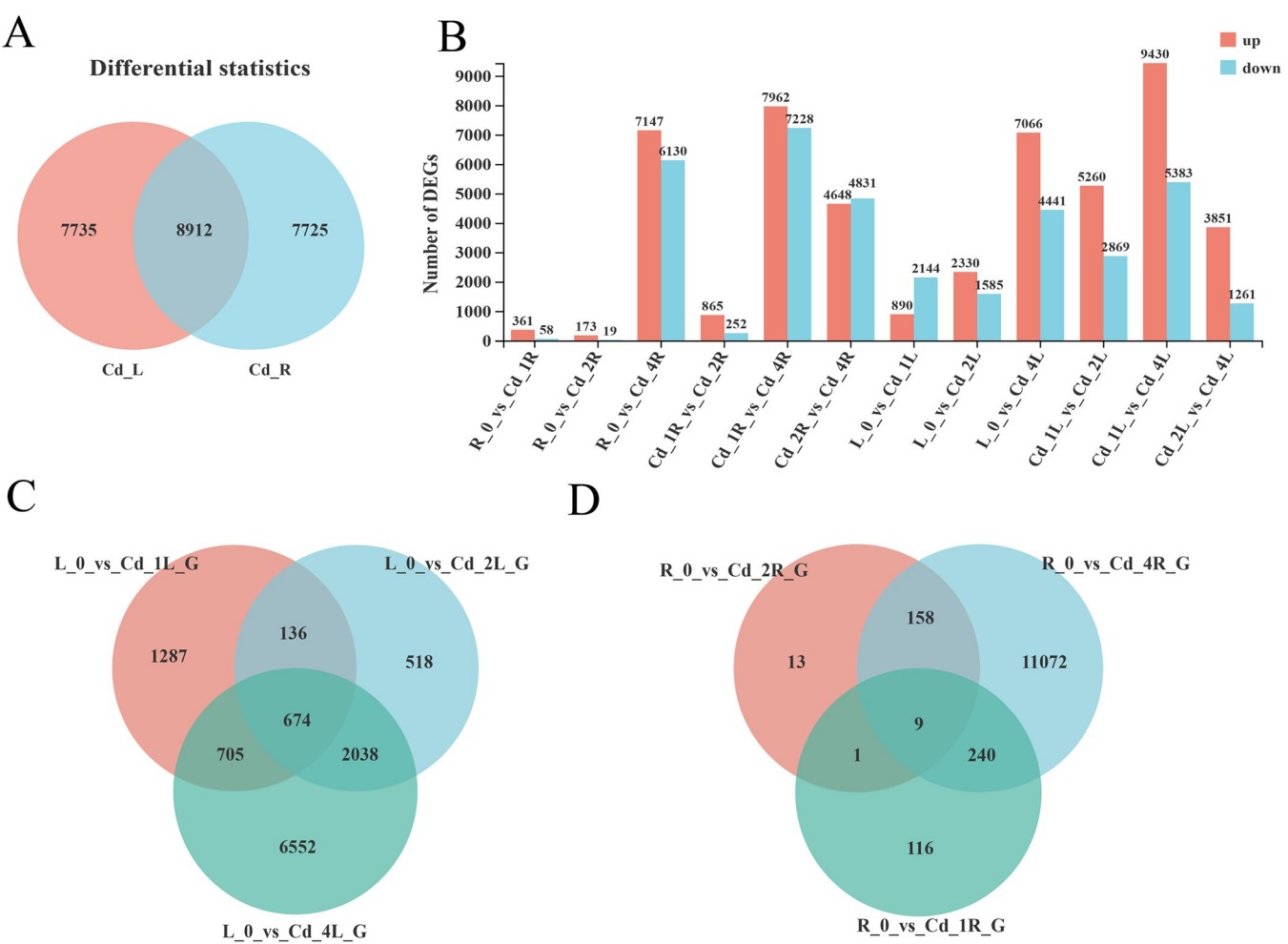

**Fig 3.** Gene expression in leaves and roots treated with Cd for 0, 1, 2, and 4 days was analyzed using RNA-seq (A) Statistics of the number of differentially expressed unigenes (DEGs) in roots and leaves. (B) The number of DEGs in different comparison groups. (C) Venn diagram of DEGs in different leaf samples. (D) Venn diagram of DEGs in different root samples. L and R are the leaves and roots, respectively.

In addition, the DEGs present in both roots and leaves were involved in the MAPK signaling pathway—plant, plant hormone signal transduction, flavonoid biosynthesis and photosynthesis—antenna proteins (Fig 5A). The DEGs unique to roots were mostly enriched in thiamine metabolism, glyoxylate and dicarboxylate, cutin, suberine and wax biosynthesis metabolism (S3A Fig). The DEGs unique to leaves were involved in the pentose phosphate pathway, fructose and mannose metabolism, glutathione metabolism, monobactam biosynthesis and glycosaminoglycan degradation (S3B Fig). Moreover, GSEA showed that the DEGs between the Cd-treated group and the control group were mainly closely related to the glutathione metabolism (map00480), photosynthesis (map00195), flavonoid biosynthesis (map00941), ABC transporter (map02010), and endocytosis (map04144) pathways, which was similar to the KEGG enrichment results (Fig 5B).

## Identification of DEGs involved in plant hormones under Cd stress

Plant hormones are crucial in plant defense responses to abiotic stresses. To determine whether plant hormones are involved in the response to Cd treatment in *M. lutarioriparia*, we

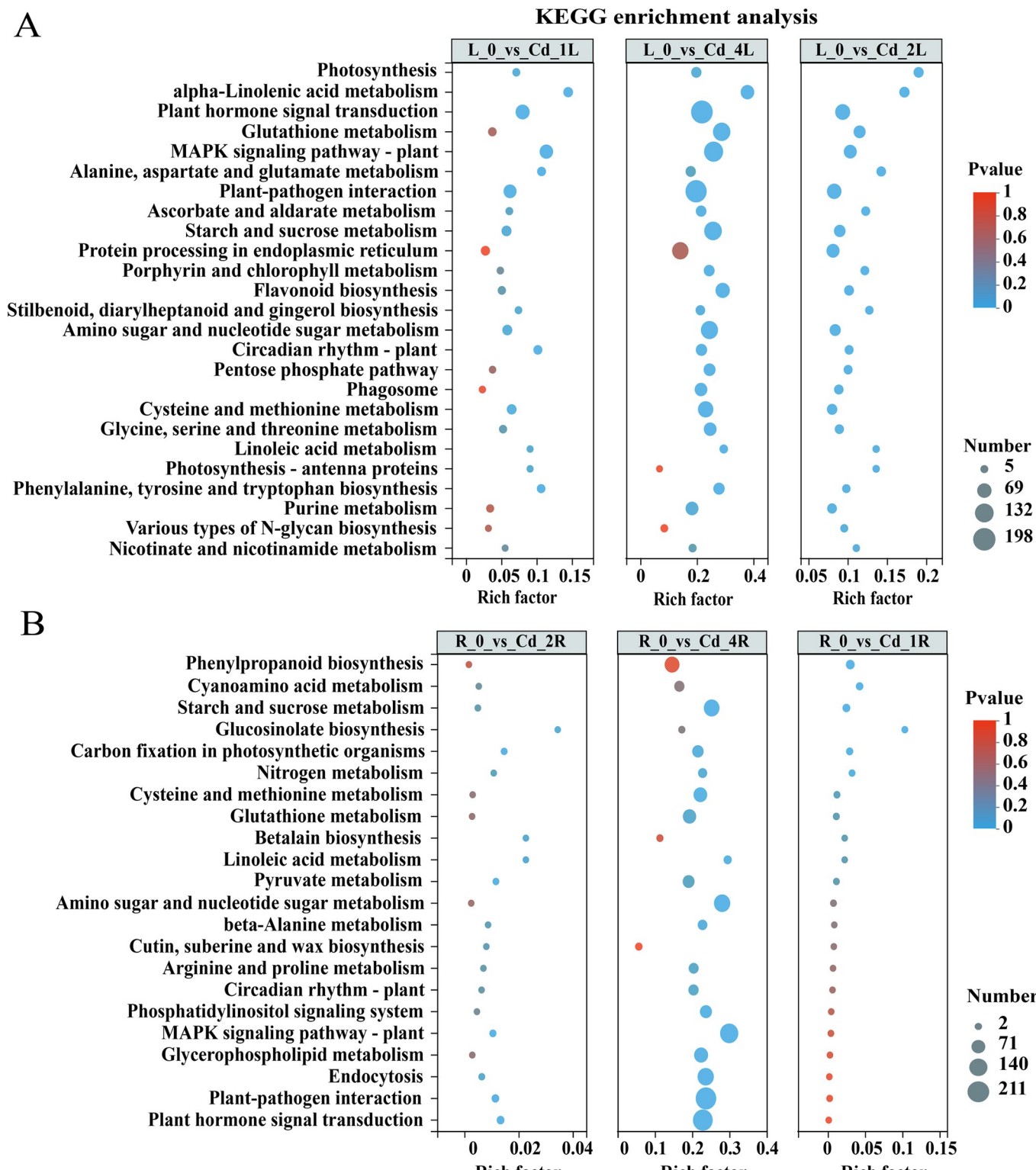

**Fig 4. KEGG enrichment analysis of DEGs in leaves and roots under the Cd stress.** (A) KEGG enrichment analysis of DEGs in leaves. (B) KEGG enrichment analysis of DEGs in roots. The x-coordinate represents the enrichment rate, and the y-coordinate represents the pathway. L and R are the leaves and roots, respectively.

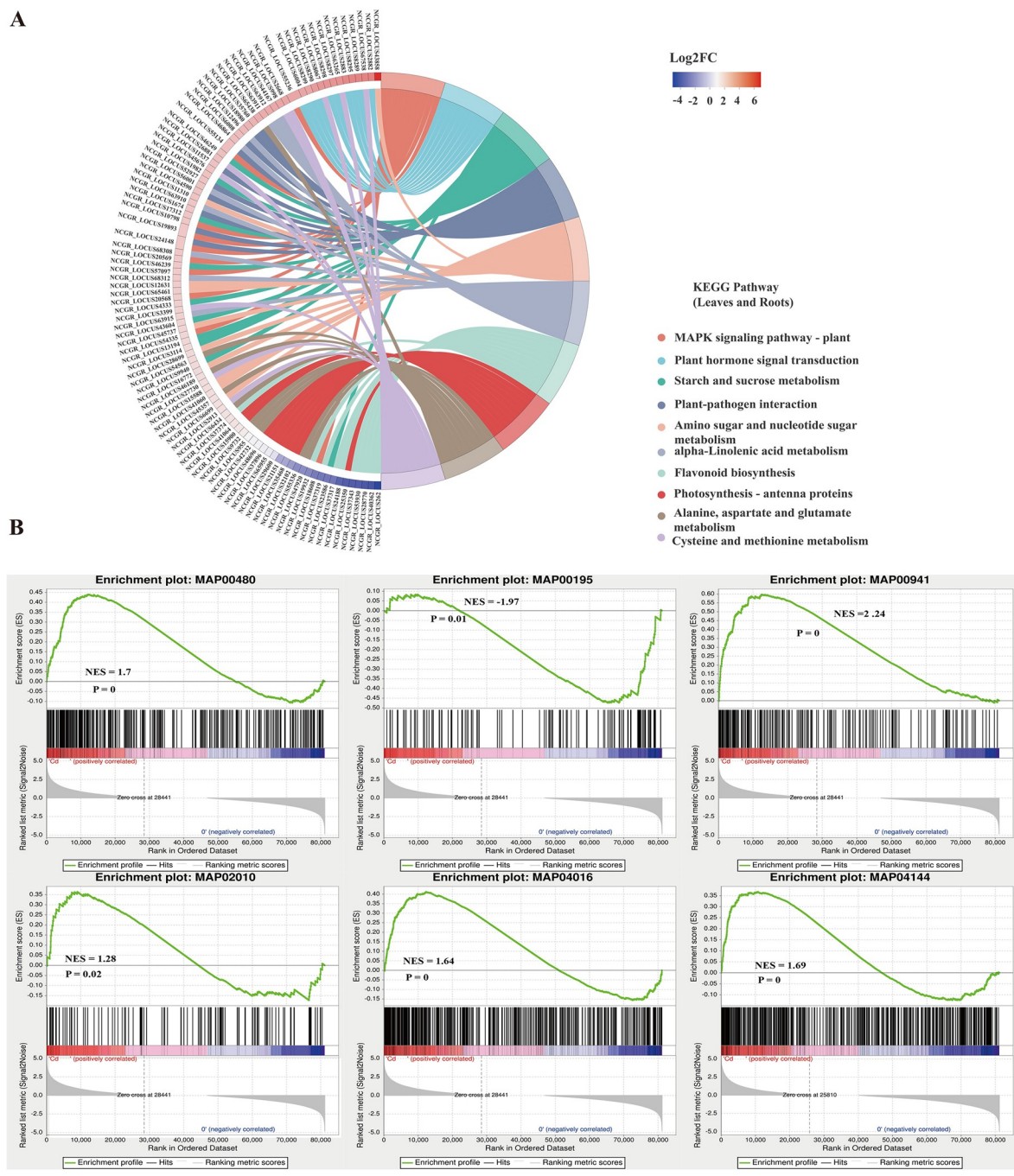

**Fig 5. Enrichment analysis of DEGs under the Cd stress.** (A) KEGG enrichment analysis of the DEGs in both roots and leaves. (B) GSEA of the DEGs between the Cd-treated group and the control group. NES: Normalized enrichment score.

analyzed the DEGs involved in plant hormone signal transduction and synthesis. In total, 168 DEGs involved in plant hormone signal transduction were identified (S4 Fig), including 34 related to auxins (IAA), 8 related to gibberellins (GA), 39 related to abscisic acid (ABA), 16 related to ethylene (ET), 16 related to brassinosteroids (Br), 37 related to jasmonates (Ja), and 8 related to cytokinins (CK). In addition, the expression levels of most genes were upregulated under Cd treatment. In addition, 46 genes related to plant hormone synthesis were identified

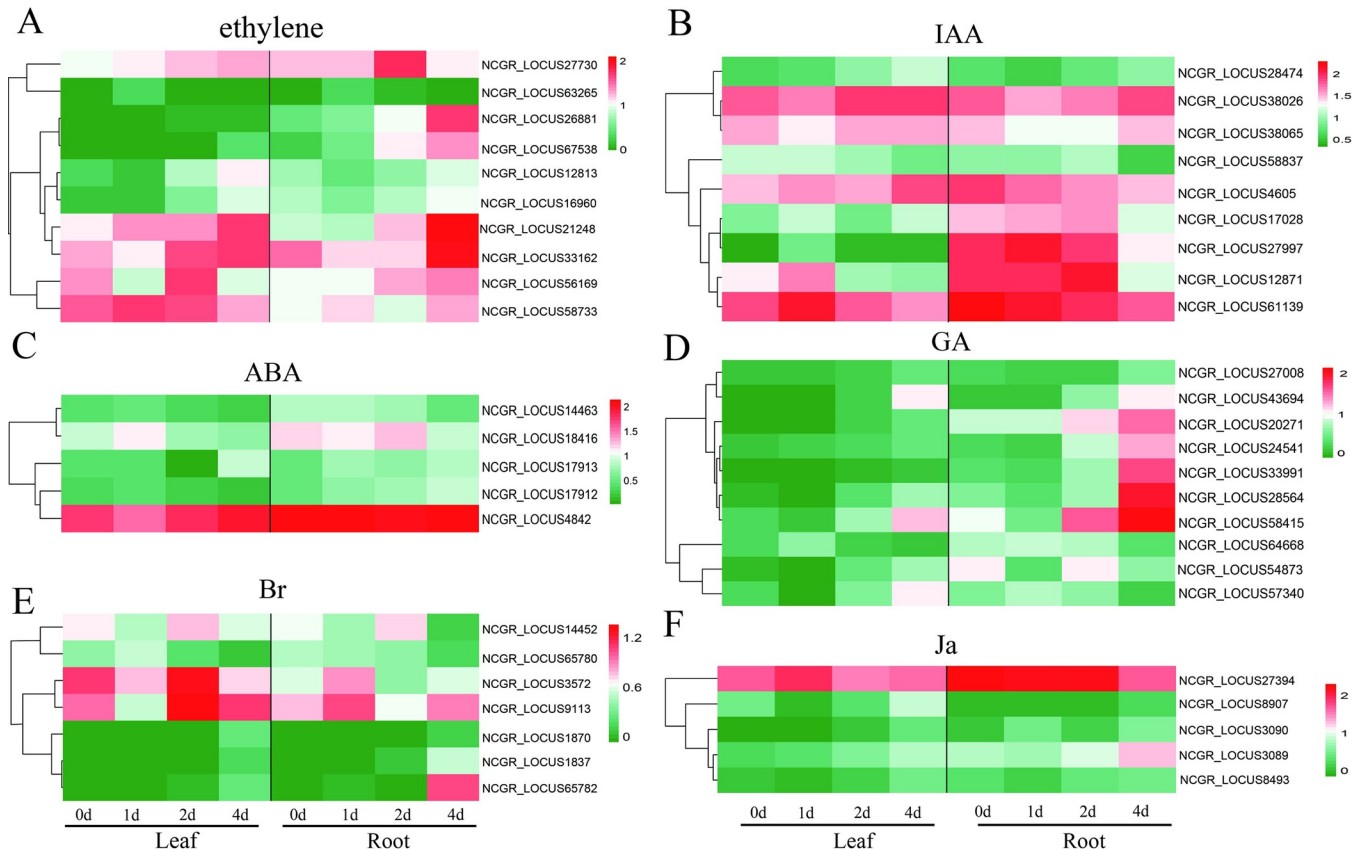

**Fig 6.** DEGs involved in plant hormone synthesis in response to Cd treatment, including ethylene (a), auxins (IAA, b), abscisic acid (ABA, c), gibberellins (GA, d), brassinosteroids (Br, e), and jasmonates (Ja, f). Red indicates upregulated DEGs, while green indicates downregulated DEGs. 0d, 1d, 2d, and 4d of plants treated with Cd for the indicated periods.

(Fig 6). Among them, 10 DEGs related to ethylene synthesis, including the 9 aminocyclopropane-1-carboxylate oxidase 1 gene and 1 ACC synthase 2 gene, were upregulated under Cd stress (Fig 6A). Nine DEGs related to IAA synthesis, including one indole-3-acetaldehyde oxidase, one indole-3-pyruvate monooxygenase, and seven aldehyde dehydrogenase genes, were upregulated, except for NCGR_LOCUS38065 (Fig 6B). Five DEGs were related to ABA synthesis, including two indole-3-acetaldehyde oxidase genes, two aldehyde oxidase genes and the zerumbone synthase gene (Fig 6C). Ten DEGs were related to GA synthesis, including 7 gibberellin 2-beta-dioxygenase genes, 2 gibberellin 2-oxidase 4 genes and one cytochrome P450 88A1 gene (Fig 6D). The expression levels of these genes continuously increased with increasing stress duration, and the highest expression level was reached on the 4th day. In addition, there were 7 and 5 genes related to Br and JA synthesis, respectively, and their expression levels also tended to increase during Cd stress (Fig 6E and 6F).

## Identification of DEGs involved in the antioxidant system under Cd stress

Under Cd stress, plants maintain the balance of ROS production and clearance through the regulation of the antioxidant system. In this study, 41 DEGs were found to be associated with antioxidant enzyme synthesis, including 5 superoxide dismutase (SOD) genes, 5 catalase (CAT) genes, 16 peroxidase (POD) genes and 15 other antioxidant enzyme-encoding genes (Fig 7). Moreover, the expression trends of different antioxidant enzyme-encoding genes were

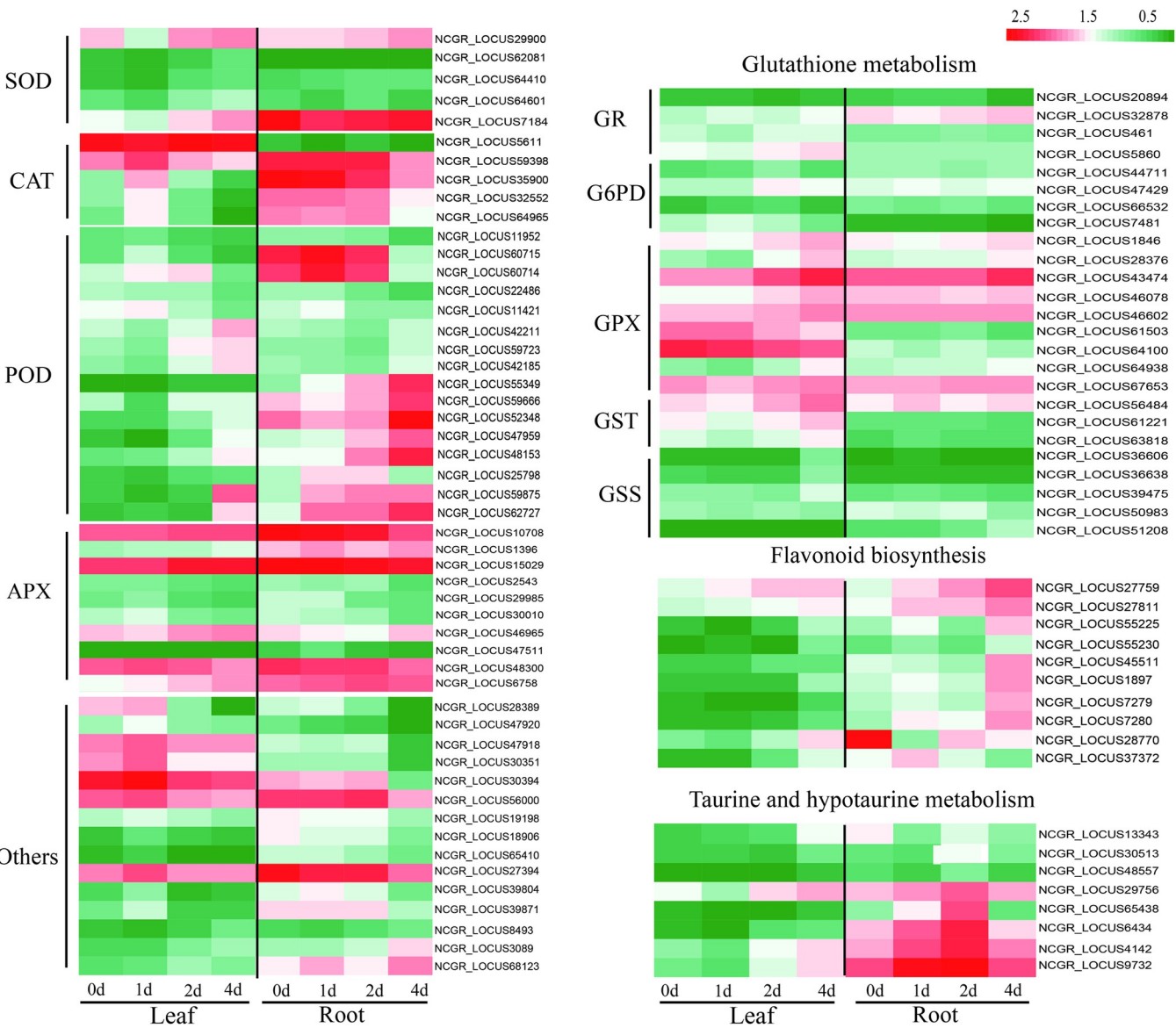

**Fig 7. Expression level analysis of the antioxidant system DEGs under Cd stress.** Notes: catalase (CAT), superoxide dismutase (SOD), peroxidase (POD), L-ascorbate peroxidase (APX), glutathione reductase (GR), glucose-6-phosphate 1-dehydrogenase (G6PD), glutathione peroxidase (GPX), glutathione S-transferase (GST), and glutathione synthetase (GSS). Red indicates upregulated DEGs, while green indicates downregulated DEGs. 0d, 1d, 2d, 4d of plants treated with Cd for the indicated periods.

different. For example, the expression of CAT-related enzyme-encoding genes was significantly upregulated on the first day of Cd stress, while the expression of SOD and POD was significantly upregulated mostly on the second and fourth days of Cd stress. In addition, 53 DEGs related to the synthesis of other antioxidants, including 35 related to glutathione metabolism, 8 related to hypotaurine metabolism and 10 related to flavonoid biosynthesis, were obtained. Among them, the DEGs related to flavonoid biosynthesis and hypotaurine metabolism were mostly upregulated on the 2nd and 4th days of Cd stress. The glutathione metabolism-related DEGs tended to increase throughout the stress period. These results indicated that on the first day of Cd treatment, the ROS balance in *M. lutarioriparia* was regulated mainly by increasing the expression levels of CAT and glutathione metabolism genes, and on

days 2 and 4, it mainly by increasing the expression of SOD, POD, flavonoid biosynthesis and flavonoid metabolism-related genes to adapt to the Cd stress.

## Response to metal ion- and transport-associated DEGs under Cd stress

To addressCd stress, plants have developed several defense mechanisms and adaptation strategies, including restriction of metal ion uptake, translocation, chelation, and compartmentalization. Here, we analyzed DEGs related to adaptation strategies, including the uptake of metal ions (76), response to metal ions (59), response to toxic substances (23), ABC transporters (20) and metal ion transmembrane activity (S5 Fig). These data revealed that the expression levels of most DEGs, including heavy metal transport/detoxification superfamily proteins, heavy metal-associated isoprenylated plant proteins (HMA), zinc transporters, cadmium/zinc-transporting ATPase HMA3, metal transporter NRAT1 (NRAMP2), heat shock 70 kDa proteins, ABC transporters (ABCC1, ABCB1, ABCG2), glutathione S-transferase (GST-IV, GST1, GST1-like), potassium transporters, copper transporter genes, etc., were upregulated under Cd stress, especially on the first day or the fourth day. Moreover, some DEGs showed increased expression in leaves but decreased expression in roots. These genes included the ZIP zinc/iron transport family protein, inactive cadmium/zinc-transporting ATPase HMA3 and oligopeptide transporter 2 genes. These results indicated that these DEGs are critical in Cd transport and detoxification in *M. lutarioriparia*.

## Identification of DEGs involved in the photosynthetic system under Cd stress

Cd is a strong photosynthesis inhibitor that mainly inhibits the photosynthetic efficiency of plants by changing leaf stomatal conductance and chlorophyll synthesis. A total of 55 DEGs related to the plant photosynthetic system were obtained, including 28 related to n photosynthesis, 18 related to photosynthesis-antenna proteins and 9 related to chlorophyll metabolism (S6 Fig). The expression levels of most photosynthesis DEGs including the PsaK, PsaL, PsaN, PsaE, PsbP, PsbQ, PsbR, and PsbY genes were downregulated in response to Cd treatment in the leaves. In contrast, DEGs related to photosynthesis-antenna proteins were significantly upregulated at the initial stage (first day), but downregulated from the second day under Cd stress in leaves. These genes included LHCA1, LHCA2, LHCA3, LHCA4, LHCB1, and LHCB16. With respect to chlorophyll metabolism, the chlorophyll synthase, STAY-GREEN, STAY-GREEN LIKE, NYC1, and chlorophyllase gene expression levels tended to decrease first and then increase during the Cd stress period. However, the expression level of chlorophyllide and oxygenase first increased and then decreased. The results showed that the increasing the expression of photosynthesis- related antenna protein genes and changing the chlorophyll content may be the major mechanisms by which the photosynthetic system adapts to Cd stress in *M. lutarioriparius*. However, the photosynthetic efficiency decreased.

## Discussion

Cd is a highly toxic pollutant in soil that is detrimental to the growth and development of crops and human health. In particular, heavy metal pollution caused by mining seriously damages the surrounding ecological environment and endangers the health of the local people. Phytoremediation is a promising technology for remediating soil Cd pollution because of its low cost, environmental friendliness and high adaptability [2,7]. *M. lutarioriparia* has a strong heavy metal tolerance and enrichment of various heavy metals, including Cd, and has been regarded as a potential crop for phytoremediation. For example, heavily soil pollution caused a high Cd concentration in the biomass, and the accumulation of Cd increased with delayed

harvest for *M. lutarioriparia* [36]. In pot experiments using soil spiked with 10 mg Cd·kg$^{-1}$, M. floridulus reduced total Cd in rhizosphere soil by 49.2% after sixty days [37]. In this study, the Cd content of soil covered by *M. lutarioriparius* was 3–6 times lower than that of bare land in a mining area. This result was similar to the findings that *M. lutarioriparius* significantly reduced the Cd content in abandoned mine soil [23]. In addition, different varieties have different Cd adsorption capacities in soils with relatively high cadmium nitrate concentrations. The highest Cd concentration in the aboveground and belowground parts of the plants were 185.7 mg kg$^{-1}$ and 186.8 mg kg$^{-1}$, respectively. These values were significantly greater than those of other Cd hyperaccumulators, such as, *Phytolacca acinosa*(14.46 mg kg$^{-1}$ in leaves and 15.53 mg kg$^{-1}$ in roots) and *Sedum plumbizincicola* (152.93 mg kg$^{-1}$ in shoots and 90.84 mg kg$^{-1}$ in roots) [38]. Moreover, the Cd content in the leaves and root tissues significantly increased under Cd stress. On the 4th day, the Cd content in the leaves increased by more than 9 times compared with that in the control group, indicating that *M. lutarioriparius* has a strong ability to enrich and transfer Cd. however,the Cd content in the roots was significantly greater than that in the leaves, indicating that the Cd taken up was mainly retained in the belowground part. Researchers have shown similar results in *M. sacchariflorus* [23]. Thus, *M. lutarioriparius* is an excellent plant that can effectively remove and remediate Cd pollution and improve the soil environment.

Cd has an obvious inhibitory effect on plant photosynthesis, which reduces the content of photosynthetic pigments, destroys the chloroplast structure, and affects the normal function of the PSII electron transport system and photochemical reactions [39,40]. For example, the chlorophyll structure in leaves was disrupted, and the contents of various photosynthetic pigments decreased under Cd stress in *Oryza sativa*, *Morus atropurpurea*, *Brassica napus* and other crops [41–43]. In this study, the total chlorophyll content in the leaves tended to increase first and then decrease, which was not consistent with the findings for other plants. For example, the total chlorophyll content was significantly reduced under Cd stress in tobacco, wheat and Arabidopsis [44,45]. This difference could be due to the temporary increase in chlorophyll caused by the decrease in chlorophyll metabolism and the constant synthesis rate (S4 Fig). This is also a possible a mechanism of Cd stress resistance in *M. lutarioriparia*. In fact, the degradation of chlorophyll is mainly achieved by regulating the activity of chlorophyllase under Cd stress [45]. The expression of the *chlorophyll synthase (CHLG)*, *chlorophyllide a oxygenase (CAO)*, *STAY-GREEN*, *and STAY-GREEN LIKE* genes was consistent with the change in chlorophyll content in *M. lutarioriparia*. In particular, in the early stage of Cd treatment, the high expression of CHLG was contrary to the findings of other studies showing that CHLG expression is inhibited [46]. In addition, the *STAY-GREEN LIKE* gene-overexpressing plants exhibited premature leaf senescence, while the leaves of the *STAY-GREEN LIKE* gene function deletion mutant sgrl-1 plants exhibited a stagnant green color, indicating that the *STAY-GREEN LIKE* gene is important role in the process of leaf senescence induced by abiotic stress [47]. In addition, the activities of PSI and PSI and the efficiency of CO2 are significantly inhibited with Cd treatment [41]. Here, the gene expression levels of the PSI protein, PSII protein and LHC protein genes involved in the light response were downregulated. This showed that Cd stress has serious toxic effects on photosynthesis, such as light absorption and electron transfer. This result was consistent with findings that photosynthesis is a sensitive biological process that occurs in response to Cd stress [48,49]. Overall, Cd stress is inducible and involves transient photosynthetic heterogeneity as an emergency defense mechanism for *M. lutarioriparia*.

In plants, Cd reduces the efficiency of nutrient and water absorption and transport, increases oxidative damage, disrupts metabolic stability, and inhibits plant growth [28,50]. Thus, plants have also developed many response mechanisms to adapt to stress, including

antioxidant defense systems, chelation of metal ion-binding proteins, regulation of plant hormones, and mechanisms by which various proteins to absorb, transport, and detoxify heavy metals [28]. The accumulation of ROS in plants is the first toxic mechanism of Cd stress, and the antioxidant system can remove excess ROS to maintain the balance of ROS and relieve Cd stress in plants [50–53]. The plant antioxidant system includes the antioxidant enzymatic system and nonenzymatic system [53,54], which include CAT, SOD, POD, dehydro-ascorbate reductase (DHAR), ascorbate peroxidase (APX), glutathione reductase (GR), carotenoids, ascorbic acid (ASA) and glutathione (GSH). APX and GR are the major enzymes in the ascorbic acid-glutathione cycle and detoxify heavy metals by regulating the redox state of cells. In this study, the CAT and SOD enzymes participated in the detoxification by increasing their activities. This result was similar to the findings in *Hibiscus cannabinus*, *Oryza sativa*, and *Morus atropurpurea*, which have been shown to increase the activity of SOD under Cd treatment [41]. In addition, the expression levels of CAT, SOD and other antioxidant enzymes (POD, APX, DHAR, GPX, and GR) were also significantly upregulated in *M. lutarioriparia*. The results of the enzyme activity and the gene transcript profile were consistent under Cd stress, which is similar to Cd stress in other plants. For instance, Cd stress induces upregulated expressions of FeSOD, MnSOD, Chl Cu/ZnSOD, Cyt Cu/ZnSOD, APX, GPX, GR and POD [55]. In transgenic Arabidopsis plants, overexpressing SaCu/Zn SOD also increased the Cd uptake capacity and oxidative stress resistance [56]. Moreover, we found that the expression of flavonoid and glutathione-related genes was upregulated, which was consistent with previous findings that Cd can increase the gene expression levels of flavonoids and glutathione-related enzymes [57]. Flavonoids and glutathione compounds can participate in the regulation of plant stress resistance as plant antioxidants [58]. In *Camellia sinensis*, the upregulation expression levels of glutathione metabolic genes contributes to protecting plants from Cd stress [48]. Exogenous GSH alleviates the toxic effect of Cd stress on maize by altering the production of ascorbate-related metabolites and flavonoid-related metabolites[58]. In addition, exogenous flavonoids increase the Cd tolerance of plants by altering the permeability of cell membranes and immobilizing excess Cd in the cell wall [59].

Plant hormones are also involved in the mechanism by which plants adapt to heavy metal stress. For example, IAA significantly alleviates Cd toxicity to restore the growth and development of plants [60]. Moreover, CK alleviates the inhibition of photosynthetic pigments and chloroplast membranes caused by Cd stress, thereby increasing plant photosynthetic capacity and primary metabolite levels. In addition, CK triggers sulfur deficiency-like gene expression accompanied by a decrease in glutathione content, whereas cytokinin-deficiency is associated with elevated glutathione content and $Cd^{2+}$ tolerance and accumulation [61,62]. ABA inhibits the absorption and accumulation of Cd in most plants. Such as, ABA induces the downregulation of the Fe transporter IRT, which can transport Cd in Arabidopsis, thereby inhibiting the uptake of Cd [63]. Here, most of the DEGs related to phytohormone biosynthesis were significantly upregulated under Cd stress. In particular, the genes related to IAA, CK, ethylene and GA synthesis included ACC synthase 2, indole-3-acetaldehyde oxidase, gibberellin 2-oxidase, SAUR32, PIF4, PP2C, MKK4, and EIN3. Similar results were found in wheat [64]. In addition, endogenous ABA is vital to Cd transport from roots to shoots under Cd stress [65]. It has been suggested that the levels and signaling pathways of growth maintenance hormones (IAAs, CKs, GAs, and ABAs) are enhanced and/or maintained under Cd treatment, which may be a strategy for plants to avoid long-term Cd toxicity [42]. In addition, we found that some genes, such as the TCH4 gene (NCGR_LOCUS59709, NCGR_LOCUS59710, and NCGR_LOCUS62590), exhibited opposite changes in the leaves and roots.The expression level first increased and then decreased in the leaves but continued to increase in the roots under Cd stress. This result indicated that there were some differences in the regulatory effects of

phytohormones on Cd stress in leaves and roots. In *Cucurbita pepo*, the indole acetic acid content increased in roots but decreased in shoots under the treatment with 100 μM Cd [66].

Limiting the adsorption of heavy metal ions and reducing their transfer to aerial parts are the major mechanisms by which plants adapt to Cd stress[8]. Plants thicken the root system by increasing the root cell wall to increase the adsorption rate and reduce the chance of heavy metals entering cells [27]. Transporter proteins, including ABC transporters, ZIP family proteins, the metal transporter ZIP family, the yellow stripe-like (YSL) transporters, and the proton-coupled metal-ion transporters (NRAMPs), are the major tools used for Cd absorption and transport in plants [67,68]. In *Arabidopsis*, the AtABCC1, AtABCC2 and AtABCC3 protein participates in the detoxification of Cd by forming complexes with plant chelating peptides [68]. OsNRAMP1 and OsNRAMP5 encode plasma membrane-localized proteins that contribute to the uptake and transport of Cd in rice [69]. Here, the transporter protein-encoding gene expression levels of HMA, zinc transporter, cadmium/zinc-transporting ATPase HMA3, metal transporter NRAT1 (*NRAMP2*), and ABC transporters (*ABCC1*, *ABCB1*, *and ABCG2*) were significantly upregulated, which indicated that high expressions of transporter protein genes are vital in Cd stress. These results are similar to studies for other crops. For example, in the Cd/Zn hyperaccumulator S. plumbizincicola, high expression of SpHMA3 is vital in Cd detoxification by sequestering Cd into vacuoles [70]. In *Spirodela polyrhiza* and *tobacco*, *SpNramp1* overexpression significantly increased the content of Cd and Mn, and the fresh weight of the plants [71]. High expression of *OsNramp5* also increased the uptake of Cd in rice grains [72]. In addition, overexpression of *MsYSL1* enhances the resistance of Arabidopsis to Cd by mediating metal ion reallocation.

Heat shock 70 kDa protein and glutathione S-transferase (*GST-IV*, *GST1*, *GST1-like*) gene expression was also upregulated, which was similar to the findings of a recent study [73]. The synthesis of heat shock protein (70 kDa phosphoprotein) significantly increased under Cd stress in plants [74]. GSTs are detoxifying enzymes that catalyze the nucleophilic attack of the sulfur atom of GSH on the electrophilic groups of the various toxic substrates [73,75]. In addition, GSTs also reduce oxidative stress by removing various ROSs [76]. Furthermore, 49 lignin synthesis-related DEGs were identified, most of which were significantly upregulated, including the *PAL, CCR, CAD, HCT, 4CL and CCoAOMT* genes (S7 Fig). In *Salix matsudana*, the expression levels of lignin biosynthesis genes were significantly upregulated in roots under Cd stress. In particular, the *SmCCR1* and SmCAD7 genes enhance Cd tolerance in transgenic poplar calli by increasing the lignin content of roots [77].

## Conclusions

In summary, the adsorption capacity of *M. lutarioriparia* for Cd in soil was greater. It significantly reduced the soil Cd content in its growth environment and can be used for bioremediation in mining areas. In addition, the greater activity of SOD and CAT was detected in the leaves and roots after a short-term exposure to Cd. The total chlorophyll content tended to increase first and thendecrease under Cd stress. Furthermore, the DEGs involved in plant hormone signal transduction, glutathione metabolism, flavonoid biosynthesis, phenylalanine metabolism, metal ion transport and response to abiotic stimuli were obtained, which revealed that these genes respond to Cd stress in *M. lutarioriparia*. These results lay a solid foundation for breeding excellent Cd-resistant *M. lutarioriparia* and identifying candidate genes for the genetic improvement of other crops species. In addition, our findings increase our understanding of the detoxification mechanism of Cd stress in *M. lutarioriparia* and provide a new candidate plant for phytoremediation.

## Supporting information

**S1 Fig. Gene expression was analyzed using RNA-seq in leaves and roots under the Cd stress.** (a) Principal components (PCA) analysis the transcriptome similarity and repeatability between samples. (b) Gene annotation. (c) The analysis of differentially expressed genes (DEGs). The red indicates up-regulated expression of DEGs, while green indicates down-regulated expres-sion of DEGs. (d) Venn diagram of DEGs in different root and leaves samples. L0, 1L, 2L, 4 L, leaves of M. lutarioriparia treated with Cd for indicated periods. R0, 1 R, 2 R, 4R, roors of M. lutarioriparia treated with Cd for indicated periods.
(TIF)

**S2 Fig. GO enrichment analysis of DEGs in leaves and roots under the Cd stress.** (a) GO enrichment analysis of DEGs in leaves, (b) GO enrichment analysis of DEGs in roots. The x-coordinate represents the enrichment rate, and the y-coordinate represents the pathway.L and R are the leaves and roots, respectively.
(TIF)

**S3 Fig. KEGG enrichment analysis of the DEGs of unique roots and unique leaves under the Cd stress.** (a)KEGG enrichment analysis of DEGs in unique leaves, (B) KEGG enrichment analysis of DEGs unique in roots.
(TIF)

**S4 Fig. Expression level analysis of plant hormone signal transduction DEGs under the Cd stress.** abscisic acid (ABA), ethylene (ET), auxins (IAA), brassino-steroids (Br), Cytokinin (CK), Salicylic acid (Sa), jasmonates (Ja), and gibberellins (GA),. The red indicates up-regulated expression of DEGs, while green indicates down-regulated expression of DEGs. 0d, 1d, 2d, 4d, days of plants treated with Cd for indicated periods.
(TIF)

**S5 Fig.** Expression level analysis of metal ion- and transport-associated DEGs under the Cd stress, including metal ion transport (a), response to metal ions (b), response to toxic substances (c), and ABC transporters (d). The red indicates up-regulated expression of DEGs, while green indicates down-regulated expression of DEGs. 0d, 1d, 2d, 4d, days of plants treated with Cd for indicated periods.
(TIF)

**S6 Fig.  Expression level analysis of photosynthetic system DEGs under the Cd stress**, including photosynthesis (a), photosynthesis-antenna proteins (b), chlorophyll metabo lism (c). The red indicates up-regulated expression of DEGs, while green indicates down-regulated expression of DEGs. 0d, 1d, 2d, 4d, days of plants treated with Cd for indicated periods.
(TIF)

**S7 Fig. Expression level analysis of lignin synthesiem DEGs under the Cd stress.** Phenylalanine ammonia-lyase (PAL), cinnamate 4 hydroxylase (C4H), 4-hydroxycinnamoyl-CoA ligase (4CL), cinnamyl alcohol dehydrogenase(CAD), cinnamoyl-CoA reductase(CCR), hydroxycinnamoyltransferase(HCT), Cytochrome P450(C3H), ferulic acid 5-hydroxylase(F5H), caffeoyl-CoA O-methyltransferase (CCoAOMT). The red indicates up-regulated expression of DEGs, while green indicates down-regulated expression of DEGs. 0d, 1d, 2d, 4d, days of plants treated with Cd for indicated periods.
(TIF)

**S1 Table. Primer sequences used in the RT-qPCR experiment.**
(XLSX)

**S2 Table. The information on the geographical location of the sampled mining area.**
(XLSX)

**S3 Table. Summary of the raw sequencing data.**
(XLSX)

**S4 Table. Gene annotation with different public protein databases.**
(XLS)

**S5 Table. The expression levels of genes in all samples.**
(XLS)

**S6 Table. Differentially expressed genes in different samples.**
(XLS)

**S7 Table. Representative genes expression validation using quantitative real-time PCR (RT-qPCR).**
(XLSX)

## Acknowledgments

We would like to thank the Key Laboratory of Industrial Dust Prevention and Control & Occupational Safety and Health of the Ministry of Education for their experimental platform and technology support. We deeply appreciate the Majorbio I-Sanger Cloud Platform for assisting with the data analysis.

## Author Contributions

**Conceptualization:** Ying Diao.

**Data curation:** Jia Wang, Feng lin Zhu.

**Formal analysis:** Xinyu Liu, Yiran Chen.

**Funding acquisition:** Jia Wang.

**Investigation:** Jiajing Sheng.

**Methodology:** Xinyu Liu, Feng lin Zhu.

**Project administration:** Yiran Chen.

**Resources:** Yiran Chen.

**Software:** Xinyu Liu.

**Visualization:** Feng lin Zhu.

**Writing – original draft:** Jia Wang, Jiajing Sheng.

**Writing – review & editing:** Jia Wang, Jiajing Sheng, Ying Diao.

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
