## [Decision Letter · Decision Letter 0]

8 Mar 2024

PONE-D-23-38398Physiological and transcriptomic analyses reveal cadmium tolerance mechanism of Miscanthus lutarioripariaPLOS ONE

Dear Dr. diao,

Thank you for submitting your manuscript to PLOS ONE. After careful consideration, we feel that it has merit but does not fully meet PLOS ONE’s publication criteria as it currently stands. Therefore, we invite you to submit a revised version of the manuscript that addresses the points raised during the review process.

Please submit your revised manuscript by Apr 22 2024 11:59PM. If you will need more time than this to complete your revisions, please reply to this message or contact the journal office at plosone@plos.org. Please include the following items when submitting your revised manuscript:A rebuttal letter that responds to each point raised by the academic editor and reviewer(s). You should upload this letter as a separate file labeled 'Response to Reviewers'.A marked-up copy of your manuscript that highlights changes made to the original version. You should upload this as a separate file labeled 'Revised Manuscript with Track Changes'.An unmarked version of your revised paper without tracked changes. You should upload this as a separate file labeled 'Manuscript'.

We look forward to receiving your revised manuscript.

Kind regards,

Anwar Hussain

Academic Editor

PLOS ONE

Journal Requirements:

Did you know that depositing data in a repository is associated with up to a 25% citation advantage (https://doi.org/10.1371/journal.pone.0230416)? If you’ve not already done so, consider depositing your raw data in a repository to ensure your work is read, appreciated and cited by the largest possible audience. You’ll also earn an Accessible Data icon on your published paper if you deposit your data in any participating repository (https://plos.org/open-science/open-data/#accessible-data).

"This research was funded by Independent Research fund of Key Laboratory of Industrial Dust Prevention and Control & Occupational Health and Safety (Anhui University of Science and Technology) (NO. EK20202002), the Independent Research fund of Joint National-Local Engineering Research Centre for Safe and Precise Coal Mining (Anhui University of Science and Technology) (NO. EC2021007) and the Postdoctoral Science Foundation of China (2019M662719)."

"We would like to thank the Key Laboratory of Industrial Dust Prevention and Control & Occupational Safety and Health of the Ministry of Education for their experimental platform and technology support. We deeply appreciate the Majorbio I-Sanger Cloud Platform for the help in data analysis. This research was funded by Independent Research fund of Key Laboratory of Industrial Dust Prevention and Control & Occupational Health and Safety (Anhui University of Science and Technology) (NO. EK20202002), the Independent Research fund of Joint National-Local Engineering Research Centre for Safe and Precise Coal Mining (Anhui University of Science and Technology) (NO. EC2021007) and the Postdoctoral Science Foundation of China (2019M662719)."

"This research was funded by Independent Research fund of Key Laboratory of Industrial Dust Prevention and Control & Occupational Health and Safety (Anhui University of Science and Technology) (NO. EK20202002), the Independent Research fund of Joint National-Local Engineering Research Centre for Safe and Precise Coal Mining (Anhui University of Science and Technology) (NO. EC2021007) and the Postdoctoral Science Foundation of China (2019M662719)."

5.Thank you for stating the following in your Competing Interests section: "NO authors have competing interests"

6. Please note that your Data Availability Statement is currently missing the direct link to access each database. If your manuscript is accepted for publication, you will be asked to provide these details on a very short timeline. We therefore suggest that you provide this information now, though we will not hold up the peer review process if you are unable.

**Additional Editor Comments:**

As per reviewers report, the study falls within the scope of Plos One and can be accepted for publication after major revision. You are advised to carefully follow the instructions of the reviewers and revise the manuscript accordingly. 

Reviewers' comments:

Reviewer's Responses to Questions

**Comments to the Author**

1. Is the manuscript technically sound, and do the data support the conclusions?

Reviewer #1: Yes

Reviewer #2: Yes

2. Has the statistical analysis been performed appropriately and rigorously? 

Reviewer #1: Yes

Reviewer #2: Yes

3. Have the authors made all data underlying the findings in their manuscript fully available?

Reviewer #1: Yes

Reviewer #2: Yes

4. Is the manuscript presented in an intelligible fashion and written in standard English?

Reviewer #1: No

Reviewer #2: Yes

5. Review Comments to the Author

Reviewer #1: Dear Authors,

A detailed report regarding the manuscript, titled "Physiological and transcriptomic analyses reveal cadmium tolerance mechanism of Miscanthus lutarioriparia" is attached for your perusal. Please answer the queries raised in the revised files.

Best wishes

Reviewer #2: The current study reports on the Physiological and transcriptomic analyses reveal cadmium tolerance mechanism of Miscanthus lutarioriparia. Overall, this is a good piece of work with interesting findings. The paper may be accepted for publication after the below minor corrections.

1. The abstract lacks clarity, specific methodological details.

2. Revise the key words.

3. Abbreviation should be in full form on its first appearance.

4. Line 177-78 to evaluate the adsorption capacity of Cd in soil for M. lutarioriparia should be deleted.

5. Line 188 to evaluate the Cd adsorption properties of M.lutarioriparia, the content of Cd in treatment group and control group should be deleted.

7. It is better to add some more explanation and references in the discussion section.

8. Some of the references are missing essential details, such as the volume and page numbers for journal articles, publication years etc.

6. PLOS authors have the option to publish the peer review history of their article (what does this mean?). If published, this will include your full peer review and any attached files.

Reviewer #1: **Yes: **Amjad Iqbal

Reviewer #2: **Yes: **Asif Mehmood

---

## [Author Response · Author response to Decision Letter 0]

28 Mar 2024

Response: We thank the Editor for the professional advices and we revised it in the new version.

Did you know that depositing data in a repository is associated with up to a 25% citation advantage (https://doi.org/10.1371/journal.pone.0230416)? If you’ve not already done so, consider depositing your raw data in a repository to ensure your work is read, appreciated and cited by the largest possible audience. You’ll also earn an Accessible Data icon on your published paper if you deposit your data in any participating repository (https://plos.org/open-science/open-data/#accessible-data).

Response: We thank the Editor for the professional advices and we revised it in the new version.

The database: https://www.ncbi.nlm.nih.gov/bioproject/?term=PRJNA733881

"This research was funded by Independent Research fund of Key Laboratory of Industrial Dust Prevention and Control & Occupational Health and Safety (Anhui University of Science and Technology) (NO. EK20202002), the Independent Research fund of Joint National-Local Engineering Research Centre for Safe and Precise Coal Mining (Anhui University of Science and Technology) (NO. EC2021007) and the Postdoctoral Science Foundation of China (2019M662719)." The funders had no role in study design, data collection and analysis, decision to publish, or preparation of the manuscript.

Response: We thank the Editor for the professional advices and we revised it in the new version.

This research was funded by Independent Research fund of Key Laboratory of Industrial Dust Prevention and Control & Occupational Health and Safety (Anhui University of Science and Technology) (NO. EK20202002), the Independent Research fund of Joint National-Local Engineering Research Centre for Safe and Precise Coal Mining (Anhui University of Science and Technology) (NO. EC2021007) and the Postdoctoral Science Foundation of China (2019M662719)." The funders had no role in study design, data collection and analysis, decision to publish, or preparation of the manuscript.

"We would like to thank the Key Laboratory of Industrial Dust Prevention and Control & Occupational Safety and Health of the Ministry of Education for their experimental platform and technology support. We deeply appreciate the Majorbio I-Sanger Cloud Platform for the help in data analysis. This research was funded by Independent Research fund of Key Laboratory of Industrial Dust Prevention and Control & Occupational Health and Safety (Anhui University of Science and Technology) (NO. EK20202002), the Independent Research fund of Joint National-Local Engineering Research Centre for Safe and Precise Coal Mining (Anhui University of Science and Technology) (NO. EC2021007) and the Postdoctoral Science Foundation of China (2019M662719)."

"This research was funded by Independent Research fund of Key Laboratory of Industrial Dust Prevention and Control & Occupational Health and Safety (Anhui University of Science and Technology) (NO. EK20202002), the Independent Research fund of Joint National-Local Engineering Research Centre for Safe and Precise Coal Mining (Anhui University of Science and Technology) (NO. EC2021007) and the Postdoctoral Science Foundation of China (2019M662719)."

Response: We thank the Editor for the professional advices and we revised it in the new version.

5. Thank you for stating the following in your Competing Interests section: "NO authors have competing interests"

Please complete your Competing Interests on the online submission form to state any Competing Interests. If you have no competing interests, please state ""The authors have declared that no competing interests exist." ", as detailed online in our guide for authors at http://journals.plos.org/plosone/s/submit-now

Response: We thank the Editor for the professional advices and we revised it in the new version.

6. Please note that your Data Availability Statement is currently missing the direct link to access each database. If your manuscript is accepted for publication, you will be asked to provide these details on a very short timeline. We therefore suggest that you provide this information now, though we will not hold up the peer review process if you are unable.

Response: The direct link of database: https://www.ncbi.nlm.nih.gov/bioproject/?term=PRJNA733881

Reviewer #1: 

Dear Authors,

I have gone through the manuscript, titled: “Physiological and transcriptomic analyses reveal cadmium tolerance mechanism of Miscanthus lutarioriparia”. The work is good, but there are numerous queries throughout the manuscript that should be answered before final decision by the journal Editor.

Response: We are very grateful to the reviewer for the professional advices. The language of this manuscript has been reedited by professional English colleagues. We supplemented related references as support of statement in the introduction and discussion. We also filled some methodological details in this new version. Other questions/concerns that highlighted on the attached document were all revised in the current version.

Abstract

The author made a general summary without giving specific details about the results. It's important for them to share the findings using percentages. Also, they should clearly explain the importance of the study in this part. The language of the manuscript is poor and need to be checked by a native speaker before submission of the revised version.

Response: We are very grateful to the reviewer for the professional advices and we revised it in the new version. We have added some detailed results. The language of this manuscript has been reedited by professional English colleagues.

Line 40-41: You conclude “These results lay a solid foundation for breeding excellent Cd resistant M. lutarioriparia and provide candidate genes for the genetic improvement of other crops.” Even if the crops accumulate more cadmium (Cd) without affecting their yield and other growth aspects; it would still be harmful to humans, isn’t it?

Response: Yes，we strongly agree with your point . But, the crops in the paper are specific plants that can be used for phytoremediation (A review on phytoremediation of contaminants in air, water and soil. Journal of hazardous materials, 2021, 403: 123658. ), not food crops that we eat. The aim of the research is to obtain a plant that can be used to repair soil heavy metal pollution.

Line 41: The word should be “candidate genes” rather “can didate genes”.

The conclusion is not very clear. Please write a clear conclusion.

Response: We thank the reviewer for pointing this out and we apologize for mischaracterizing it in our manuscript. 

Introduction

In the introduction section, it is important to first state the hypothesis behind the study, outlining the expected outcomes. Following this, the specific objectives of the study should be clearly written.

Response: We are very grateful to the reviewer for the professional advices and we revised it in the new version.

Line 70 & 72: You wrote “Miscanthus×giganteus” any specific reason? 

Response: We are very grateful to the reviewer for the professional advices. The Miscanthus×giganteus is a triploid hybrid of a diploid Miscanthus sinensis and a tetraploid Miscanthus sacchariflorus. So, Miscanthus×giganteus is Latin name. 

Such as the reference: Evaluation of Miscanthus× Giganteus Tolerance to Trace Element Stress: Field Experiment with Soils Possessing Gradient Cd, Pb, and Zn Concentrations. Plants, 2023,12(7), 1560. Remediation of soils on municipal rendering plant territories using Miscanthus× giganteus. Environmental Science and Pollution Research, 2023, 30(9): 22305-22318.

Please include more recent references in the list if possible, avoiding those that are over 10 years. old 

Response: We are very grateful to the reviewer for the professional advices and we revised it in the new version. 

M&M

It's important to know if the soil used in the experiments was sterile.

Response: The soil is not sterile. These soil from the field was air-dried and sieved over a 4 mm mesh before used.

Where you kept the pots during the experiment?

Response: The pots were kept in glasshouses (day/night temperature 28℃/25℃, light/darkness duration 16 h/8 h, and humidity 60%) during the experiment.

Line 108: “Each treatment was planted with 3 pots, and each pot was planted with 3 plants.” The sentence is so confusing. What do you mean?

Response: We are very sorry for the confusion caused to you by our writing. This means “all experimental materials are planted in treated pots, and each material is not less than 9 plants”

Line 109: You used the word “natural growth”, what do you mean?

Response: We are very sorry for the confusion caused to you by our writing. This means “Plants grow normally in glasshouses (day/night temperature 28℃/25℃, light/darkness duration 16 h/8 h, and humidity 60%)” 

Line 111: You checked physiological and transcriptomic effects of M. lutarioriparia (Ml004) leaves and roots. Why you excluded stem?

Response: We thank the reviewer for the professional advices. Because the root is the first or direct part of the plant to contact Cd, and the leaf is the last part to contact Cd (after the root absorbs it, it is finally transported to the leaf). So, we mainly focused physiological and transcriptional changes in leaf and root tissues. 

What was the frequency of irrigation?

Response: Rinse twice with tap water and three times with sterile water.

Results

You have mentioned that the cadmium concentration of the soil was increased to 150 mg/kg and the plants ended up with 185.65±15.54 mg/kg (aboveground parts) and 186.80±14.97 mg/kg (belowground parts), explain please. 

Response: The dry weight of each plant is about 100 g in the experiment, and the total amount of Cd adsorbed is 18.6 mg based on the concentration of 186 mg /kg. Each pot contained 3kg of soil, the Cd content was 450 mg based on the concentration of 150 mg /kg. So, the plant adsorb only 4.1% of the Cd content in the soil.

In hydroponic experiment you have given 18 mg/L of Cd stress, while you ended up 40 mg/kg of Cd in leaves and approximately 100 mg/kg in roots, explain please.

Response: The dry weight of each plant is about 10 g in the experiment, and the total amount of Cd adsorbed is 0.4-1 mg based on the concentration of 40-100 mg /kg. The content of Cd in each culture solution (2L) was 36 mg based on the concentration of 18 mg /L. So, the plant adsorb only 1.1-2.7% of the Cd content in the culture solution.

Line 189: I reckon it is better to divide the aboveground parts in to leaves and shoot to get a better view of the Cd accumulation.

Response: We thank the reviewer for the professional advices and we strongly agree with your point of view. But, in this study, we mainly considered to calculate the Cd transfer rate of plants (TF = (Cd concentration in aboveground part)/(Cd concentration in belowground part). So, we mixed leaves and stems as above-ground parts for detection. In the future, we will further analyze the Cd content of the leaves, above-ground stems, underground stems and root tissues of Miscanthus lutarioriparia according to the ideas provided from you. 

Discussion

Please add those references that are not 10-years old. 

Response: We thank the reviewer for the professional advices and we revised it in the new version.

Line 355, 360 & 361: You wrote “mgl-1, mgkg-1”, it should be written as “mg.l-1, mg.kg-1”. Check throughout and correct accordingly.

Response: We thank the reviewer for pointing this out and we apologize for mischaracterizing it in our manuscript. We reedited the reference in the new version.

Overall

The language of the manuscript is not up to the mark. At many instances the sentences are confusing and should be rewritten. It would be helpful to have a native English speaker or a skilled writer reviews the document before submitting the revised version.

Response: We are very grateful to the reviewer for the professional advices. The language of this manuscript has been reedited by professional English colleagues.

Reviewer #2: The current study reports on the Physiological and transcriptomic analyses reveal cadmium tolerance mechanism of Miscanthus lutarioriparia. Overall, this is a good piece of work with interesting findings. The paper may be accepted for publication after the below minor corrections.

Response: We are very grateful to the reviewer for the professional advices. The language of this manuscript has been reedited by professional English colleagues. We supplemented related references as support of statement in the introduction and discussion. We also filled some methodological details in this new version. Other questions/concerns that highlighted on the attached document were all revised in the current version.

1. The abstract lacks clarity, specific methodological details.

Response: We thank the reviewer for the professional advices and we revised it in the new version. We filled some methodological details in this new version.

2. Revise the key words.

Response: We thank the reviewer for the professional advices and we revised it in the new version.

3. Abbreviation should be in full form on its first appearance.

Response: We thank the reviewer for the professional advices and we revised it in the new version.

4. Line 177-78 to evaluate the adsorption capacity of Cd in soil for M. lutarioriparia should be deleted.

Response: We thank the reviewer for the professional advices and we revised it in the new version.

5. Line 188 to evaluate the Cd adsorption properties of M.lutarioriparia, the content of Cd in treatment group and control group should be deleted.

Response: We thank the reviewer for the professional advices and we revised it in the new version.

6. It is better to add some more explanation and references in the discussion section.

Response: We are very grateful to the reviewer for the professional advices. We supplemented related references as support of statement in the discussion.

8. Some of the references are missing essential details, such as the volume and page numbers for journal articles, publication years etc.

Response: We thank the reviewer for pointing this out and we apologize for mischaracterizing it in our manuscript. We reedited the reference in the new version.

---

## [Decision Letter · Decision Letter 1]

16 Apr 2024

Physiological and transcriptomic analyses reveal the cadmium tolerance mechanism of Miscanthus lutarioriparia

PONE-D-23-38398R1

Dear Dr. diao,

We’re pleased to inform you that your manuscript has been judged scientifically suitable for publication and will be formally accepted for publication once it meets all outstanding technical requirements.

Kind regards,

Anwar Hussain

Academic Editor

PLOS ONE

Additional Editor Comments (optional):

Reviewers' comments:

Reviewer's Responses to Questions

**Comments to the Author**

1. If the authors have adequately addressed your comments raised in a previous round of review and you feel that this manuscript is now acceptable for publication, you may indicate that here to bypass the “Comments to the Author” section, enter your conflict of interest statement in the “Confidential to Editor” section, and submit your "Accept" recommendation.

Reviewer #1: All comments have been addressed

Reviewer #2: All comments have been addressed

2. Is the manuscript technically sound, and do the data support the conclusions?

Reviewer #1: Yes

Reviewer #2: Yes

3. Has the statistical analysis been performed appropriately and rigorously? 

Reviewer #1: Yes

Reviewer #2: Yes

4. Have the authors made all data underlying the findings in their manuscript fully available?

Reviewer #1: Yes

Reviewer #2: Yes

5. Is the manuscript presented in an intelligible fashion and written in standard English?

Reviewer #1: Yes

Reviewer #2: Yes

6. Review Comments to the Author

Reviewer #1: Dear Authors,

I have gone through the revised version of your paper and found that all my queries are well addressed. Therefore. I am please to recommend the acceptance of your paper.

Best wishes

Reviewer #2: (No Response)

7. PLOS authors have the option to publish the peer review history of their article (what does this mean?). If published, this will include your full peer review and any attached files.

Reviewer #1: **Yes: **Prof. Amjad Iqbal

Reviewer #2: **Yes: **Asif Mehmood

---

## [Editor Report · Acceptance letter]

3 May 2024

PONE-D-23-38398R1 

PLOS ONE

Dear Dr. diao, 

I'm pleased to inform you that your manuscript has been deemed suitable for publication in PLOS ONE. Congratulations! Your manuscript is now being handed over to our production team.

Kind regards, 

on behalf of

Dr. Anwar Hussain 

Academic Editor

PLOS ONE